# Microbiome–Gut–Mucosal–Immune–Brain Axis and Autism Spectrum Disorder (ASD): A Novel Proposal of the Role of the Gut Microbiome in ASD Aetiology

**DOI:** 10.3390/bs13070548

**Published:** 2023-06-30

**Authors:** Amapola De Sales-Millán, José Félix Aguirre-Garrido, Rina María González-Cervantes, José Antonio Velázquez-Aragón

**Affiliations:** 1División de Ciencias Biológicas y de la Salud, Universidad Autónoma Metropolitana-Lerma, Lerma 52006, Estado de Mexico, Mexico; 2212803115@correo.ler.uam.mx; 2Departamento de Ciencias Ambientales, Universidad Autónoma Metropolitana-Lerma, Lerma 52006, Estado de Mexico, Mexico; j.aguirre@correo.ler.uam.mx (J.F.A.-G.); rgonzalez@correo.ler.uam.mx (R.M.G.-C.); 3Experimental Oncology Laboratory, National Pediatric Institute, Mexico City 04530, Mexico

**Keywords:** microbiome-gut-brain axis, autism spectrum disorder, dysbiosis, gastrointestinal functions, immune system, and neuroimmunogastroenterology

## Abstract

Autism Spectrum Disorder (ASD) is a complex neurodevelopmental disorder characterised by deficits in social interaction and communication, as well as restricted and stereotyped interests. Due of the high prevalence of gastrointestinal disorders in individuals with ASD, researchers have investigated the gut microbiota as a potential contributor to its aetiology. The relationship between the microbiome, gut, and brain (microbiome–gut–brain axis) has been acknowledged as a key factor in modulating brain function and social behaviour, but its connection to the aetiology of ASD is not well understood. Recently, there has been increasing attention on the relationship between the immune system, gastrointestinal disorders and neurological issues in ASD, particularly in relation to the loss of specific species or a decrease in microbial diversity. It focuses on how gut microbiota dysbiosis can affect gut permeability, immune function and microbiota metabolites in ASD. However, a very complete study suggests that dysbiosis is a consequence of the disease and that it has practically no effect on autistic manifestations. This is a review of the relationship between the immune system, microbial diversity and the microbiome–gut–brain axis in the development of autistic symptoms severity and a proposal of a novel role of gut microbiome in ASD, where dysbiosis is a consequence of ASD-related behaviour and where dysbiosis in turn accentuates the autistic manifestations of the patients via the microbiome–gut–brain axis in a feedback circuit.

## 1. Introduction

Autism spectrum disorder (ASD) refers to a complex neurodevelopmental disorder characterised by deficits in social interaction and communication and restricted and stereotyped behaviours, interests or activities [1]. ASD has a significant financial and health impact on individuals with autism, their families and society [2]. The cause of ASD is not yet fully understood, but research suggests that a combination of genetic and environmental factors are involved [3]. Different reports propose that, in addition to the core symptoms of ASD, gastrointestinal symptoms, including constipation, abdominal pain, vomiting, diarrhoea and gas, are frequent in people with ASD, with estimates ranging from 9% to 70% [4]. The gut microbiome is a crucial component of human physiology [5], and the frequent gastrointestinal symptoms in individuals with ASD have led to the investigation of a possible connection between the gut microbiota and the symptoms of the disorder [4]. The mode of birth and feeding type during the first years of life can condition changes in the gut microbiota. Children born vaginally and fed with breastmilk present more *Bifidobacteria* than children born by caesarean section and fed with formula. In the complementary feeding stage, the diversity of foods can determine the conformation of different microbial groups that will be maintained in the adult stage. A long-term fibre-deficient diet can lead to changes in the composition of gut microbiota (dysbiosis). These changes cause alterations in the production of important neurotransmitters for neurodevelopment [5]. There is increasing evidence supporting the importance of gut microbiota in brain development and function, showing communication along the gut–brain axis [6]. During the first 3 years of life, the consolidation of the microbiota–gut–brain axis is of vital importance because during this stage, the central nervous system (CNS) develops rapidly [7], and gut microbiota also suffer important changes due to changes in feeding patterns during this developmental stage [8,9]. Recently, the focus has been expanded to the role of the gut microbiota [10], generating increased interest with findings suggesting specific microorganisms associated with memory, stress, mood, neurodevelopment and even ASD [11,12,13].

The gut–microbiome–brain axis is a multidisciplinary research area that has attracted the attention of different researchers from around the world. Multiple studies have found evidence that the gut–microbiome–brain axis is important for the mental and cognitive health of children with ASD [14,15,16]. Despite this, there have been relatively few clinical studies in humans that provide clear evidence of the role in the aetiology of neurodevelopmental disorders [17,18]. On the other hand, a paper published in 2021 by Yap et al. [19] establishes that the microbiota does not participate in autistic manifestations and that gut dysbiosis is a consequence of central symptoms in ASD.

Understanding the role of the gut–microbiome–brain axis in the development of ASD is crucial to determining its potential as a therapeutic target. This is particularly important because faecal transplants and pre/probiotic interventions have already been used to treat patients with ASD.

## 2. Gut Microbiota

The human gut microbiota is composed of 10^13^ to 10^14^ microorganisms, mostly bacteria [5]. Their collective genome is defined as the gut microbiome [20]. The number of microorganisms varies throughout the intestinal tract, with 95% of symbiotic microorganisms (the largest microecosystem in the human body) residing in the intestine [21]. They are represented by anaerobic bacteria belonging, in order of abundance, to two phyla: Bacteroidetes and Firmicutes, accounting for approximately 90% of the bacterial species [22]. In addition to bacteria belonging to the Proteobacteria, Actinobacteria and Verrucomicrobia phyla, the gut microbiota also includes viruses, fungi and archaea (mainly in pathological conditions), maintaining a symbiotic relationship with the host [23].

The resident microbial community of the human gut has a unique combination of different types and numbers of bacteria that are constantly evolving and can be influenced by diet, environmental factors, drug treatments and age, and between many other factors. For example, the western-style diet (high fat/low fibre) is related to the decrease in *Bacteroides*, *Bifidobacterium*, and *Akkermansia* genus, affecting intestinal microbiota homeostasis and permeability. Furthermore, the western diet is characterised by an increased Firmicutes/Bacteroidetes ratio and weight gain. A relationship between obesity and the risk of infection by the enteric pathogen *Clostridium difficile* have been identified [24]. Antibiotic exposure during infancy is associated with adiposity and obesity later in childhood, which may occur through disrupted microbial colonisation that can have consequent effects on gastrointestinal motility, secretion and immune functions. Polypharmacy in older hospitalised patients is associated with dysbiosis in contrast to no dysbiosis in active healthy older people without polypharmacy. In terms of lifestyle, changes in living conditions and dietary changes have been shown to significantly influence the elderly gut microbiota [25]. In healthy individuals, the effect of geographic distance has an impact on gut microbiota variations, demonstrating the cumulative effect of dietary, genetic, lifestyle and climate factors on microbiota variations among healthy individuals [26]. In the European Metagenomics of the Human Intestinal Tract (MetaHit) project, it was possible to identify and characterise three different groups of microbiota or enterotypes [27]. Intercontinentally, enterotype 1 was dominated by the *Bifidobacterium* genus, enterotype 2 was represented by the *Prevotella* genus and enterotype 3 was represented by the *Bacteroides* genus [26].

The Firmicutes/Bacteroidetes ratio changes with age, with a ratio of approximately 0.4 in infants and up to 10.9 in adults [28]. Imbalances in their abundance cause dysbiosis [29,30,31,32,33]. Dysbiosis is generally associated with pathological conditions such as obesity, type 2 diabetes, cancer, gastrointestinal diseases, neuropsychiatric disorders such as depression and anxiety, as well as neurodevelopmental diseases such as ASD [5,21]. However, the composition and functional characteristics of a healthy microbiome remain to be defined.

### 2.1. Functions of the Gut Microbiota

The symbiotic relationship between humans and the microbiota has allowed the development of characteristics that play a fundamental role in biological processes, such as nutrient usage, host metabolism, protection against infections and maturation of the immune system [34,35]; Brain development and behaviour are also part of this symbiotic relationship, as the gut microbiota interacts with the neuroendocrine, neuroimmune and autonomic nervous systems (ANS) [5].

The gut microbiota is considered as one of the key elements that contributes to the regulation of host health. In healthy patients, the gut microbiota tends towards homeostasis by resisting change during ecological stress (resistance) or returning to a state of equilibrium after a stress-related disturbance (resilience) [36]. Numerous molecular mechanisms explain how the intestinal microbiota may be causally related to the protection or appearance of disease [21]. For example, celiac disease and inflammatory bowel disorder in infants are related to an increase in several bacterial species, such as *Dialister invisus*, *Parabacteroides spp*. or *Lachnospiraceae,* and certain metabolites, such as tryptophan, before the disease appears. Conversely, several anti-inflammatory strains, such as *Faecalibacterium prausnitzii* and *Clostridium clostridioforme*, are decreased [37].

The routes of communication between the microbiota and the brain are constantly being discovered, including the vagus nerve, intestinal hormone signalling, the immune system and tryptophan metabolites, or through bacterial metabolites such as short-chain fatty acids (SCFA), which they include propionate, butyrate and acetate [38]. Butyrate is an epigenetic modulator that acts through histone deacetylases [39]. The gut microbiota also regulates central neurotransmitters by altering the levels of their precursors in ASD, depression and anxiety, particular in neurodevelopmental disorders such as ASD (Table 1). 

### 2.2. Factors Influencing the Gut Microbiota

The transmission of strains (*Prevotella copri*) from mother to child can occur vertically in the first days of life, but the acquisition of strains from a shared environment over time cannot be ruled out, producing host-dependent co-diversification patterns [45]. During the developmental stages of the new-born, from birth to complementary feeding, the complexity and richness of the microbial community fluctuates [46]. At 12 months of age, infants’ gut microbiota is more like their mother’s, showing an increase in alpha diversity and a decrease in beta diversity as a function of time, indicating a more complex and less heterogeneous community [47].

The maturation of the microbial ecosystem depends on the coexistence with family members [48]. There are different related factors that, with individuality, have a crucial role in shaping the microbial composition of the intestine [49]. These factors include age, host genetics, antibiotic use, colonisation site physiology, mode of delivery (vaginal section or caesarean section), a type of feeding (human milk or formula) and environment of birth (rural or urban) [50,51,52,53].

The colonisation process during the first days of the new-born’s life is critical for correct immunological, cognitive and physiological development [54,55,56]. Caesarean section delivery with antibiotic treatment before prophylaxis is related to a decrease in the diversity of Actinobacteria, Bacteroidetes, *Bifidobacterium*, and *Lactobacillus*, and an increase in Proteobacteria, Firmicutes, and *Enterococcus* spp. On the other hand, vaginal delivery with the administration of antibiotics before delivery is related to a decrease in *Bifidobacterium* and an increase in *Clostridium* [57]. The effect of birth by caesarean section on the intestinal microbiota normalises at the age of 3 to 5 years; however, the intestinal microbiota does not reach the complexity of an adult at 5 years of age, since the infant’s gut microbiota is different to his mother and different to adults [46,47]. This indicates that both, the composition (dynamic) and structure of the gut microbiota evolve as infants grow up in response to environmental factors and the diet that feed the microbial community [47]. 

The gut microbiota of babies at 4 and 12 months is more heterogeneous and different to an adult but reaches a greater similarity at the age of 3 and 5 years [47,53]. At 4 months of age, the most abundant bacterial genus is *Bifidobacterium* together with lactic acid bacteria (*Enterococcus*, *Streptococcus* and *Lactobacillus*) and Proteobacteria (*Enterobacteriaceae*, *Citrobacter* and *Serratia*) [47]. Some adults may have lower alpha diversity and community type than children, which suggests that they have an immature microbiota, identifying *Bifidobacterium* and *Ruminococcus gnavus* as markers of immature microbiota for both children and adults [47,57]. *R. gnavus* can use the host’s mucin and is linked to inflammatory problems of the gastrointestinal tract [46].

### 2.3. Changes in the Composition of the Gut Microbiota

The main inhabitants of the intestine in a vaginally born baby approximately 1 month after birth are Actinobacteria (*Bifidobacterium*), Bacteroidetes (*Bacteroides*), and Gammaproteobacteria (*Enterobacteria*), which are essential for the metabolism of specific carbohydrates in breast milk [25,58]. At around the 6 months of life, with the introduction of a solid diet, the infant’s microbiota experiences change in relative abundance, with a decrease in *Bifidobacterium* and *Clostridial*, and an increase in Firmicutes [25]. On the other hand, caesarean section delivery is related to a decrease in *Bacteroides*, with a 41% similarity to the maternal microbiota compared to vaginal delivery [52,54,59].

Caesarean section delivery and formula feeding could be related to an increase in Bacteroides, Proteobacteria and Firmicutes, compared to vaginally born, breastfed children, who show higher dominance of *Bifidobacterium* and *Bacteroides* dominance [59,60]. Breast milk provides the infant with bioactive molecules, such as human milk oligosaccharides, IgA and essential fatty acids, and contributes to the optimal configuration of the gut microbiota, as well as influencing cognitive development and the maturation of the immune system [25].

Throughout life, various environmental factors can alter the gut microbiota [54], but the first 3 years of life are particularly important for the health of the host. However, the microbiota continues to change over the years, becoming more diverse and showing a gradual increase in the proportion of *Bacteroides* and *Clostridium* under normal conditions [59]. When there is an imbalance in microbial abundance, dysbiosis occurs. This imbalance may result from a loss of microbial diversity, either from a loss of beneficial microorganisms or an increase of pathogenic species, which is often linked to pathological states, such as some neurobehavioral and gastrointestinal problems, that are more commonly seen in children with ASD [30,61]. Gastrointestinal problems in children with ASD are mainly due to the increase in *Bacteroidetes*, *Clostridium* and *Suterella* [18,62,63].

### 2.4. Dysbiosis and Gastrointestinal Function in ASD

The prevalence of gastrointestinal symptoms in children with ASD ranges from 9% to 91%; for this reason, the study of the gut microbiota has been a key point in the exploration of gastrointestinal disorders and their relationship with the development of different ASD phenotypes [64]. The severity of the symptoms in ASD, the form of birth and maternal age are factors that affect the diversity of the intestinal microbiota. A lower abundance of *Bacteroides* and *Faecalibacterium* and higher abundance of *Clostridial* has been found in stool samples from children with autism, and higher levels of *Erysipelotrichaceae* and *Faecalibacterium* were also identified in children with severe autism [31].

Under normal circumstances, the intestinal microbiota is composed of anaerobic bacteria that are essential for protecting against pathogens and performing various functions that benefit the host, such as nutrient absorption, immunity and production of short-chain fatty acids, vitamins and amino acid synthesis among others, which help keep the colon healthy [5]. However, under dysbiosis, the absorption of nutrients in the small intestine is low, and therefore, more monosaccharides and disaccharides reach the large intestine, benefiting the bacteria that ferment simple sugars and show greater growth compared to bacteria that degrade complex sugars (polysaccharides). More sugars in the gut can cause gas and bloating [65]. Patients with ASD have increased intestinal permeability, which may be related to changes in microbial diversity and a decrease in the amount of *Lactobacillus*, bacteria related to maintaining the union of the epithelial barrier of the intestine [40,66].

Increased intestinal permeability increases circulating bacterial-derived lipopolysaccharides (LPS) that trigger immunological and inflammatory reactions, characterised by a systemic increase in proinflammatory cytokines. The increase in cytokines has been reported in patients with ASD, particularly in those subtypes of mental regression. Cytokines are necessary for normal neurodevelopment, and their disturbances can impact this process [5].

Although the results on the relationship of the neurotransmitter γ-aminobutyric acid (GABA) and the severity of autistic symptoms (Table 1) are not entirely conclusive, the reduction in GABA could be accentuating the neuroinflammatory processes caused by the increase in LPS in ASD. GABA and glutamate are the main excitatory and inhibitory neurotransmitters in the brain, and their balanced interaction is necessary for neuronal function [67]. A recent study found that GABA alleviates sepsis induced by LPS. Therefore, it is possible to relate the decrease in GABA in patients with autism to the attempt to mediate neuroinflammatory processes in these patients [68].

## 3. Gut Microbiota and the Immune System

At birth, the new-born’s immune system is not well developed, which gives it the benefit of not generating a severe reaction against the mother’s antigens [69]. Different studies suggest that the gut microbiota plays an important role as a source of antigens, including peptidoglycans, lipoproteins, lipopolysaccharides and flagellin. All these antigens together constitute, activate and educate the innate and adaptive immune systems [53,70]. A stable microbial composition (homeostasis), as well as a suitable microenvironment (as the body environment affects microbial diversity), is important for healthy metabolic functioning of microbiota. Therefore, the first colonising microorganisms in the human intestine are of vital importance [71,72].

*Bifidobacterium* is transmitted vertically from mother to child and are the first to colonise and constitute the most abundant group of microorganisms in the intestine of a healthy new-born [58,73,74]. The colonisation and establishment of this bacterial group in the intestine of the new-born have been related to the modulation by oligosaccharides, specific nutrients that human milk contains according to a microbe–host coevolution mechanism [59,75]. Different functions with health benefits, modulation of the immune system and production of metabolites that confer physiological benefits have been proposed [76,77].

Gut bacteria can act as modulators of the immune system. Extracellular structures produced by gut bacteria have been hypothesised to interact with the host’s immune system, potentially leading to pro-inflammatory or anti-inflammatory effects [76]. Extracellular structures such as pili/fimbria, exopolysaccharides and teichoic acids play a crucial role between bacterial communication and the host’s immune system. Especially during the first years of life, which are characterised by an immature immune system [73,74].

Considering the *Bifidobacterium* dominance is common until weaning, it has been suggested that these bacteria, followed by *Lactobacillus* and *Veillonella* [78], have a fundamental role in the formation of the host’s immune system [58,76]. Early childhood represents a very critical window of time for the formation of human health with lasting effects, especially in those individuals born by caesarean section or with antibiotic treatment who may suffer from altered gut microbiota [77]. Intervention with *Bifidobacterium*-based therapies can be considered a highly suitable approach to promote and maintain a balanced infant gut microbiota [74,77].

### 3.1. Inflammation and the Gut Microbiota

Epidermal growth factor is present in amniotic fluid and, after birth, in colostrum and mature milk. In the infant intestine, it promotes the proliferation and maturation of epithelial cells, which are involved in repairing the intestinal mucosa [54]. The epithelial barrier has the function of maintaining intestinal homeostasis between luminal microbes and the host’s immune system. In addition, it is the first site exposed to different environmental factors that can lead to the development of a pathological state [79]. A deterioration in the epithelial barrier can lead to the translocation of intestinal microbes, thus promoting hyperactivation of the mucosal immune system and increased production of proinflammatory cytokines, which together promote inflammation [80].

Inflammation can induce goblet cell depletion, leading to inappropriate mucin secretion and imbalanced production of IL-17, which together lead to chronic inflammation [80]. The intestinal microbiota is closely related to IL-17 levels, and dysbiosis can cause chronic inflammation. In hosts genetically susceptible to inflammation, the immune response is dysregulated towards intestinal commensal microorganisms (pathobionts). Hyperactivation of IL-17 can lead to autoimmune-type inflammation in the gut [80,81].

Because genes affect the proper functioning of the immune system and increase susceptibility to disease, certain alleles have been associated with specific types of microbial composition. The rs651821 variant of the *APOA5* gene is related to the presence of *Lactobacillus*, *Suturella* and *Methanobrevibacter* in individuals who are at a higher risk of developing metabolic diseases. Additionally, variants in the *SLIT3* gene have been associated with the development of inflammation induced by microbial products [82]. However, it is worth nothing that various studies have shown that the composition of the gut microbiota is primarily influenced by non-genetic factors [83].

### 3.2. Other Immune Alterations Related to the Gut Microbiota

The host microbiota is an important environmental factor that can confer maturation and activation to microglia, both in healthy and pathological conditions. During the prenatal stage, some maternal conditions, such as viral or bacterial infections, can activate the maternal immune system (MIA), which can then be passed on to the progeny and cause lasting changes in behaviour [84]. Although the main function of microglia is rapid protection for the brain, dysfunction of immune activity can lead to negative outcomes for surrounding neurons and glia, affecting neurogenesis, synapse and neuroinflammatory regulation [84,85]. Given that microglia cells are long-lived and part of the early brain structure, microglia with aberrant function during neurodevelopment in infancy could play a key role in modulating cell–cell interactions during life, early life and the trajectory of health and disease [86].

Gastrointestinal physiology is controlled by the enteric nervous system (ENS), which is composed of neurones and glial cells. The gut microbiota and the postnatal mucosal immune system are responsible for conferring maturation on gut neural networks [87]. Molecular interactions between the microbiota, enteric cells and immune cells (including microglia) are crucial to maintaining gastrointestinal homeostasis. A disruption in these interactions can lead to neurodevelopmental disorders, including ASD [88,89,90].

## 4. The Microbiota–Gut–Brain Axis

The gut microbiota plays a fundamental role in the physiological functioning of the host and alterations in this microenvironment can have harmful effects on key points in the development of various organs systems, including the brain and digestive system (brain–gut–microbiota or microbiota–gut–brain axis). The brain–gut axis consists of the brain, the spinal cord, ANS, ENS and the hypothalamic–pituitary–adrenal (HPA) axis. Disturbances in the microbiota–gut–brain axis are the principal cause of the most frequent gastrointestinal motility disorders [91]. Studies in germ-free (GF) animals, those treated with antimicrobials or those exposed to environmental modifications that alter the gut microbiota from the prenatal or postnatal stage have been related to problems in brain immunity, blood–brain barrier (BBB) permeability, brain architecture and neural circuits [92].

### 4.1. Animal Models of Altered Gut Microbiota and Effects in the CNS

There are bacterial strains such as *Escherichia coli* or *Lactobacillus* sp. that interact directly with the host’s CNS through neurotransmitters dopamine, norepinephrine, histamine, acetylcholine, GABA or serotonin. An alteration in the composition of these strains can lead to an alteration in the metabolic state of the microbiome, resulting in metabolic disorders that may be responsible for the severity or progression of neurological disorders, such as Parkinson’s, Alzheimer’s, ASD and depression, among others [93,94].

Experimental studies have shown that the production of the bacterial metabolite 4-methylphenol (para-cresol or *p*-cresol) can alter the composition of the intestinal microbiota, leading to the recolonisation of *Clostridium difficile*. Fermenting tyrosine via the p-hydroxyphenylacetate (p-HPA) pathway is how *C. difficile* produces *p*-cresol. Since *C. difficile* is related to decreased growth of Proteobacteria, experimental studies showed that there were animals infected with *C. difficile* mutants (hpdC), since they found bacterial families belonging to Proteobacteria. In addition, the studies found that *Bifidobacterium adolescentis* is more sensitive to the presence of *p*-cresol than other Gram-positive species [95,96].

*p*-cresol negatively affects the homeostasis of epithelial cells, and its excess negatively affects the integrity of colonic epithelial cells [97]. As a result of the disruption in epithelial cells, a proinflammatory phenotype involving LPS may be promoted. Inflammation is closely related to the pathophysiology of mental disorders. Multiple communication pathways between the microbiota and the CNS have been identified, including immune pathways [98].

### 4.2. Effects of Gut Microbiota Metabolites in Immune Cells of the CNS

The gut microbiota has a complex and specific communication system with the CNS. The communication between the microbiota, the gut and the brain involves the secretion of different metabolites, including short-chain fatty acids (SCFA), the structural components of bacteria and signalling molecules [99].

Bacterial metabolites such as LPS can easily cross the intestinal barrier and cause inflammation, which affects the brain by altering cytokine levels [100]. Additionally, cytokines produced locally in the gastrointestinal mucosa travel peripherally and can cross the BBB [101]. During inflammation, the brain releases arginine vasopressin, a metabolite that affects social behaviour and is a considered biomarker in ASD [14]. There is a bidirectional relationship between the gut and the brain that includes nerve fibres [88]. Enteroendocrine cells of the intestinal epithelial barrier can detect the composition of the intestinal lumen, as well as nutrients and bacterial metabolites. These cells synapse with afferent fibres that directly connect the intestinal lumen to the brainstem [102].

The SCFA are the result of the fermentation of dietary fibre by anaerobic commensal bacteria in the colon. The host recognises SCFA (acetate, propionate and butyrate). Recently, butyrate has been identified as protective of the mucosa via goblet cells, as they regulate the response to the upregulation of *MUC* gene expression [16,103,104,105].

Altered concentrations of metabolites may have functional consequences in ASD. Different studies have identified various altered metabolites in the urinary profile of children with autism, some even correlate with the severity of autistic behaviour such as *p*-cresol [32,106]. Additionally, the metabolic pathways for tryptophan, vitamin B6, purine and phenylalanine are altered in ASD [107,108].

Combination therapy with vancomycin and *Bifidobacterium* improved in autistic symptoms. Additionally, the above therapy helped normalise the levels of the metabolites 3-(3-hydroxyphenyl)-3-hydroxypropionic acid, 3-hydroxyphenylacetic acid and 3-hydroxyhippuric acid in the urine of children with ASD, indicating an alteration in the production of phenylalanine in ASD [109].

## 5. Gut Microbiome, Immune System and Neurodevelopment Disorders

In mammals, enteric neurogenesis and gliogenesis occur primarily during the embryonic and foetal stages, but a considerable fraction of enteric neurons and glia are born in the colonising postnatal gut [87]. Functional maturation of gut neural networks is completed within the microenvironment of the postnatal gut, under the influence of gut microbiota and the mucosal immune system [88].

The gut microbiome lies at the intersection between the environment and the host, with the ability to modify host responses to disease-relevant exposures and stimuli. This is evident in the way that enteric microbes interact with the immune system, for example, by supporting immune maturation in the first years of life, affecting the efficacy of drugs through modulation of immune responses, or influencing the development of immune cell populations and their mediators [110].

### 5.1. Altered Gut Microbiota and Neurodevelopment Disorders

Gut microbiota alterations have been linked to various pathogenic pathways, and an increasing number of studies are linking changes in the gut microbiota to a range of neuropsychiatric diseases [111,112]. Similarly, the decrease in microbial diversity throughout life could be related to neurodegeneration [113,114].

The neuroinflammation produced by the different metabolites of a dysbiotic microbiota can be a pathogenic factor in severe neurodegenerative disorders [115]. In a severe inflammatory state, the activation of microglia releases proinflammatory cytokines such as TNF-α, IL-6 or MCP-1, or the inflammasome, as well as reactive oxygen species from microglial cells and resident macrophages, which could cause chronic neuroinflammation [17,116,117]. Therefore, the decrease or loss of the integrity of intestinal epithelial cells and chronic inflammation are highlighted as the main consequences of dysbiosis. Neuroinflammation and neurodegenerative and neuropsychiatric disorders can be the result of dysbiosis [38,111,118].

CNS inflammation is related to ASD. TNF-α is found at high levels in children with ASD, which was correlated with the severity of gastrointestinal symptoms [119]. Increases in TNF-α (increase *Lachnoclostridium bolteae*), IL-2, IL-4, IL-6 (increase *Clostridium lituseburense*), IL-8, IL-10 and IL-17 (increase *Clostridium tertium*) indicate a higher level of inflammation in the CNS of children with autism [18,41,100]. Additionally, children with ASD also have dysregulated T-cell production, leading to biases in the Th1 to Th2 ratio and immune cell activation associated with altered behaviour due to further neurodevelopmental impairment [120].

### 5.2. Altered Gut Microbiota and ASD

Microbial colonisation of the gastrointestinal tract begins prenatally, as microorganisms are detected in the placenta and meconium [121,122]. A recent study found that mothers of children with ASD harbour altered gut microbiomes [123], supporting the idea that maternal gut microbiota variation and infections during pregnancy may increase the risk of ASD in offspring [15]. In this same study, a clear relationship was found between the gut microbiome profiles of children and their mothers.

Children with ASD have unique bacterial biomarkers [123]. Previous studies have suggested that the gut microbiome of children with ASD contains harmful genera or species that contribute to the severity of autism symptoms, such as *Bacteroides* [124,125] or *Desulfovibrio*, which is related to the modulation of *p*-cresol production [32]. Changes in the gut microenvironment caused by the gut microbiome affect the production of signalling substances, leading to inadequate functioning of the brain and thus the prenatal and postnatal CNS [11].

The most common gut microbiota findings in children with ASD were a decreasing trend from Bacteroidetes to Firmicutes [42] and increased abundance of *Clostridium* [29,31,126]. Despite these findings, there are inconsistencies about the phenotypic signature of the gut microbiome of children with ASD, and a reason for these inconsistent results is that the composition of the gut microbiota is influenced by several factors, such as diet, lifestyle and medical history, among others [124]. In this regard, the use of antibiotics showed a correlation with the improvement of symptoms in ASD [127]. Although recently, it has been seen this is not always the case since the prolonged use of oral antibiotics can increase the proliferation of anaerobic bacteria in the intestine. For example, *Clostridia*, *Bacteroidetes* and *Desulfovibrio* are common bacteria that, in addition to modulating the intestinal immune system, can promote gastrointestinal symptoms and autistic behaviour in ASD [128].

The fungi *Candida* also appears to play a role in children with ASD [42]. In a dysbiotic environment, as often seen in the population with autism, *Candida* proliferates and produces ammonia and toxins, which increases autistic behaviour. *Candida* also causes malabsorption of minerals and carbohydrates that play an important role in the pathophysiology of ASD [129]. A subset of people with ASD shows gastrointestinal disturbances [130], and results from different studies indicate that eliminating some foods from the diet may help improve gastrointestinal symptoms in ASD. Associating diet is an important factor in the composition of the gut microbiota [131].

On the other hand, in a study with a large group of ASD patients, their gut metagenome showed a relationship with diet, reduced taxonomic diversity and stool consistency, but no relationship was found between the diagnosis of ASD and the gut microbiome. It was suggested that softer faecal consistency is more closely related to decreased taxonomic diversity and that there is a downstream relationship to reduced dietary diversity, which is a common feature of patients with ASD. In this work, authors proposed that this mechanism could explain the relationship between the increase in gastrointestinal problems and increase in repetitive behaviours in ASD. Maintain that sensory sensitivity could be the basis of restricted diets in ASD but found no relationship between the sensory profile and ASD severity. The study found that all psychometric characteristics had more significant correlations with dietary diversity than with taxonomic diversity. In conclusion, the results suggest that dysbiosis is a consequence of autistic manifestations and that it has no causal role in the disease. Therefore, microbiome-directed treatment is not a suitable therapeutic target for treating comorbidities in children with ASD [19].

The cause of ASD remains undetermined, complex and incompletely understood, with increasing evidence pointing to abnormal synaptic development and aberrant immune responses as possible effectors of autistic symptoms [84,132]. Microglial cells have been strongly associated with physiological processes and the development of autistic symptoms, as they are part of the main cells of the CNS that provide an innate immune response to the tissue with inflammatory and tissue repair functions [84].

Some metabolites produced by specific bacterial groups have anti-inflammatory effects on microglia, such as butyrate, an essential SCFA for the modulation of excitatory and inhibitory neuronal pathways in ASD [99,133]. Patients with ASD showed low levels of SCFA [134]. The deficiency of these metabolites could be the cause of a disruption in the intercommunication between the ENS and the mucosal immune system [88], what could be generate changes in the intestinal motility of children with ASD [27]. For example, SCFA activate G protein-coupled receptors (GPR41 and GPR43) on enteroendocrine cells of the intestinal epithelium, resulting in increased production of GLP-1 and 5-HT and changes in intestinal motility [88].

Not studies that suggest that dysbiosis has a leading role in the cause of ASD, since it has always been presented as a multifactorial disease with a very high genetic component, which seems to be the main cause of autistic symptoms [123,135,136,137,138,139]. Although, dysbiosis has been proposed as another factor in the cause of ASD, its relevance has not been well clarified since the results of various studies have not been conclusive. Some studies point to dysbiosis as a possible factor in autistic symptoms [29,62], and other studies rule out the possibility that dysbiosis is a determining factor in the aetiology of ASD [19,140].

Recently, there are increased reports of evidence regarding the possible involvement of intestinal dysbiosis (Table 1) as an aetiologic factor in ASD with moderate effects. The relevance of dysbiosis as a factor in the aetiology of ASD relies on the fact that it is modifiable. There are reports of interventions that have had beneficial but modest effects on autistic manifestations. For example, faecal transplant therapies and probiotic supplementation. A pioneer group in faecal transplantation in children with ASD found that after ten weeks of intervention, there was a reduction of almost 80% in gastrointestinal symptoms and improvement in behavioural symptoms, and that these improvements persisted after eight weeks of treatment. In addition, there were beneficial changes in the abundance of *Bifidubacterium*, *Prevotella* and *Desulfovibrio*; these changes also persisted after the suspension of the intervention and two years later [141,142]. Regarding probiotics, a successful study that supplemented with probiotics children with ASD for 6 months found positive effects on some gastrointestinal symptoms, adaptive functions and sensory profiles versus the placebo group [143]. Another study found that after supplementation with probiotics and prebiotics, there was an increase in beneficial bacteria (*Bifidobacterial* and *B. longum*) and suppression of pathogenic bacteria (*Clostridium*) with a significant reduction in the severity of autistic and gastrointestinal symptoms [144].

ASD can be syndromic or non-syndromic. Syndromic ASD is often associated with chromosomal abnormalities or monogenic alterations such as Rett and Fragile X syndrome among other syndromes. Non-syndromic ASD is a complex and multifactorial condition influenced by both genetic and environmental factors [121,134,135,136,137,138]. It is widely recognised that the causative traits of ASD predominantly have a genetic basis. A meta-analysis of 13 twin studies estimated the heritability of ASD to be 74%, with an environmental effect of 25% [145]. However, it is important to acknowledge that the heritability value may be prone to overestimation due to statistical artifacts or an underestimation of unaccounted shared environmental factors, which cannot be accurately estimated in twin and sibling studies. Another estimation of ASD heritability based on 192 twin pairs found a lower value of 38% for heritability and a 58% contribution from environmental factors [146].

The contribution of genetic and environmental factors to the presentation of ASD can vary, but genetic factors are considered to be the primary determinants. Approximately 1000 human loci have been associated with ASD according to SFARI database [147]. Additionally, a meta-analysis of genome-wide association studies (GWAS) involving over 16,000 individuals identified chromosomal regions 10q24.32, 3p13, 3p25 and 8p11.23 that are associated with ASD. It is worth noting that the 10q24.32 region is also linked to schizophrenia [148].

In terms of environmental factors, advanced parental age [149], prenatal exposure to antibiotics or certain drugs [150], maternal immune activation [151], imbalances in prenatal micronutrients [152], epilepsy [153,154] and maternal obesity [155] have been shown to be associated with an increased risk of ASD. Therefore, these factors are considered as potential environmental drivers in the aetiology of ASD.

While there is a wealth of evidence suggesting the potential involvement of the gut microbiota in the manifestation of core ASD symptoms, it is important to note that the precise role of gut microbiota in the pathogenesis of ASD remains unclear. Numerous studies have focused on animal models, such as germ-free mice, while others have analysed dysbiosis in ASD patients and its impact on immunoregulators, neurotransmitters and relevant metabolite production in ASD [96,104,105,106]. However, despite the available information, no study to date has definitively established the intestinal microbiota as a causative environmental factor in ASD. Nevertheless, it has been a therapeutic target in various interventions aimed at managing ASD symptoms in patients.

The relevance of dysbiosis as a factor in the aetiology of ASD relies on the fact that it is modifiable. There are reports of interventions that have had beneficial but modest effects on autistic manifestations. For example, faecal transplant therapies and probiotic supplementation. A pioneer group in faecal transplantation in children with ASD found that after ten weeks of intervention, there was a reduction of almost 80% in gastrointestinal symptoms and improvement in behavioural symptoms, and that these improvements persisted after eight weeks of treatment. In addition, there were beneficial changes in the abundance of *Bifidubacterium*, *Prevotella* and *Desulfovibrio*; these changes also persisted after the suspension of the intervention and two years later [141,142]. Regarding probiotics, a successful study that supplemented with probiotics children with ASD for 6 months found positive effects on some gastrointestinal symptoms, adaptive functions and sensory profiles versus the placebo group [143]. Another study found that after supplementation with probiotics and prebiotics, there was an increase in beneficial bacteria (*Bifidubacterium* and *B. longum*) and suppression of pathogenic bacteria (*Clostridium*) with a significant reduction in the severity of autistic and gastrointestinal symptoms [144].

Dysbiosis has been proposed as a potential contributing factor in the development of ASD. However, its relevance in ASD aetiology remains unclear due to inconclusive results from various studies. Some studies suggest a possible association between dysbiosis and autistic symptoms [31,64], while others dismiss the notion that dysbiosis plays a decisive role in the development of ASD [19,140].

Yap and colleagues proposed an alternative perspective, suggesting that gut dysbiosis observed in ASD patients could be a consequence of the restrictive and behavioural patterns inherent to ASD. They argue that alterations in eating patterns, rather than dysbiosis, might have a minor or non-causal role in ASD. This study highlights the unconvincing evidence supporting the participation of gut microbiota in ASD. Many studies lacked statistical power due to small sample sizes and failed to consider important confounders such as sex, stool consistency and diet [19].

In this review, we propose a novel model that incorporates the role of gut microbiota in ASD while considering the findings presented by Yap and colleagues aims to explain the limited but evident effects of microbiota intervention on ASD core symptoms in patients (Figure 1).

ASD is a neurodevelopmental disorder of multifactorial aetiology where the main factors are genetic variations interacting with environmental factors (dysbiosis). (1) Genetics factors. The genetic predisposition considered as the main cause of ASD. Where most genes are related to synaptogenesis and many others with metabolic abnormalities and immune response. (2) Dysbiotic microbiome factors. Approximately 70% of the population with ASD present gastrointestinal problems that, together with a restricted diet (diet reduced in fruits and vegetables and high in fat and sugar), exacerbate the imbalance in the microbiome composition that may altered homeostasis and causing dysbiotic state, potentially triggering inflammatory responses and dysregulation on the production of neurotransmitters that control the microbiome–gut–brain axis. Therefore, a vicious cycle is generated until an equilibrium is reached in the severity of ASD manifestations that gives rise to the different phenotypes in ASD.

According to our model, the primary driving factors in the aetiology of ASD are genetic and environmental, with no direct involvement of gut microbiota in the manifestation of ASD. However, we propose that the restricted feeding patterns observed in ASD patients lead to decreased food diversity, subsequently resulting in a reduction of microbiota diversity and the development of gut dysbiosis.

The dysbiosis of the gut microbiota affects the gut–brain axis through mechanisms discussed in detail in this review (Figure 2). Therefore, the severity of autistic manifestations is accentuated. Within our proposed model, gut dysbiosis acts as a secondary factor that exacerbates the core symptoms of ASD and contributes to the occurrence of gastrointestinal manifestations until a certain threshold of severity is reached.

By incorporating the role of gut dysbiosis in accentuating ASD symptoms, our model provides a comprehensive framework that integrates genetic and environmental factors with the influence of the gut microbiota. This holistic perspective allows for a better understanding of the complex interactions underlying ASD pathogenesis and highlights the potential importance of interventions targeting gut dysbiosis in the management of ASD symptoms.

Therefore, based on the proposed model (Figure 1 and Figure 2), intervening in the vicious cycle by restoring the gut microbiota, specifically by increasing the abundance of beneficial bacteria, in individuals with ASD may lead to a modest improvement in autistic manifestations, ultimately reaching the severity level predetermined by the underlying etiologic environmental and genetic factors of ASD. Additionally, such interventions have been reported to alleviate gastrointestinal manifestations, as demonstrated in intervention studies [143,144].

By targeting the dysbiosis of the gut microbiota and promoting a healthier microbial composition, interventions hold the potential to positively influence ASD symptoms. However, it is important to note that the extent of improvement may be limited due to the dominant influence of genetic and environmental factors on the overall severity of ASD.

These findings suggest that therapeutic approaches aimed at restoring a balanced gut microbiota profile may provide some benefits for individuals with ASD, particularly in terms of gastrointestinal symptoms. Nonetheless, it is crucial to recognise that microbiota-based interventions should be considered as complementary to comprehensive treatment strategies that address the multifaceted nature of ASD. Further research is warranted to explore the optimal interventions and long-term effects on the gut–brain axis in individuals with ASD.

## 6. Conclusions

The mode of delivery at birth (caesarean section or vaginal) will determine the type of microbiota that will colonise the intestine of the new-born. The colonisation process during the first days of life helps configure a correct immunological and cognitive development in the neonate. Subsequently, the type of diet (breast milk or milk formula) will shape the alpha diversity of the infant. During the first months of life, alpha diversity is low, and its diversity increases gradually when solid foods are introduced to the baby for the first time (approximately 6 months).

During a disease process, the composition, from a dynamic and functional viewpoint, of the microbiome can vary. The changes in this dynamic system can generate the deregulation of the production of microbial metabolites, generating an imbalance, as reported under the term “dysbiosis”. Gastrointestinal problems are mostly linked to intestinal dysbiosis.

Alterations in the microbiota–gut–brain axis are the main cause of the most frequent gastrointestinal motility disorders in psychiatric disorders, such as depression and anxiety and, neurodevelopmental diseases, such as ASD. Neuroinflammation produced by the different metabolites of a dysbiotic microbiota may be a pathogenic factor in severe neurodegenerative disorders, such as Alzheimer’s disease and Parkinson’s disease. Children with ASD have unique bacterial biomarkers, and *Bacteroides* may contribute to the severity of autism symptoms. Despite these findings, there are inconsistencies regarding the phenotypic signature of the gut microbiome of children with ASD, and a reason for these inconsistent results is that the composition of the microbiome is influenced by several factors, such as diet, lifestyle, medical history and genetic variations. For this reason, there are controversies between the microbiome and the restrictive diet as possible causes of gastrointestinal comorbidities and autistic symptoms in ASD. However, with the model (Figure 2) proposed in this review, it is intended to expand knowledge of the role of gut microbiota in the patients with ASD.

The intercommunication between immune-mediated neurological and gastrointestinal disorders, leading to the proposition of a novel microbiome–gut–mucosa–immuno–brain axis, termed neuroimmunogastroenterology, in individuals with ASD. We postulate that this axis plays a significant role in a vicious cycle that contributes to the exacerbation of autistic manifestations. To gain insights into this phenomenon, large-scale clinical cohort studies involving early diagnosed ASD patients are essential to assess the potential prevention of dysbiosis and avoidance of accentuated autistic behaviour. Identifying therapeutic interventions to restore a healthy gut microbiome, thereby alleviating autistic manifestations, is paramount. Strategies, such as diversifying food intake with sensory feeding therapy, along with targeted administration of development stage-specific pre and probiotics to restore beneficial intestinal bacteria, are potential approaches. Furthermore, exploring the promising long-term positive effects of faecal microbiome transplantation is warranted.

## Figures and Tables

**Figure 1 behavsci-13-00548-f001:**
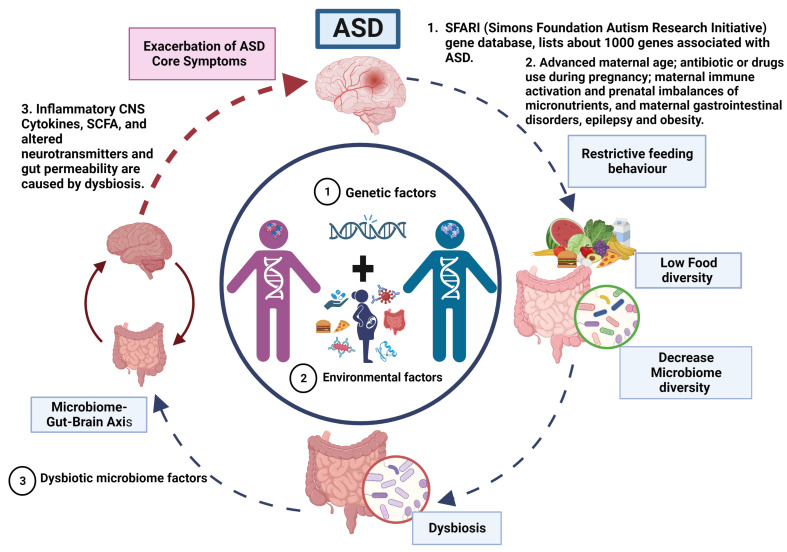
Vicious cycle and phenotype relationship in ASD.

**Figure 2 behavsci-13-00548-f002:**
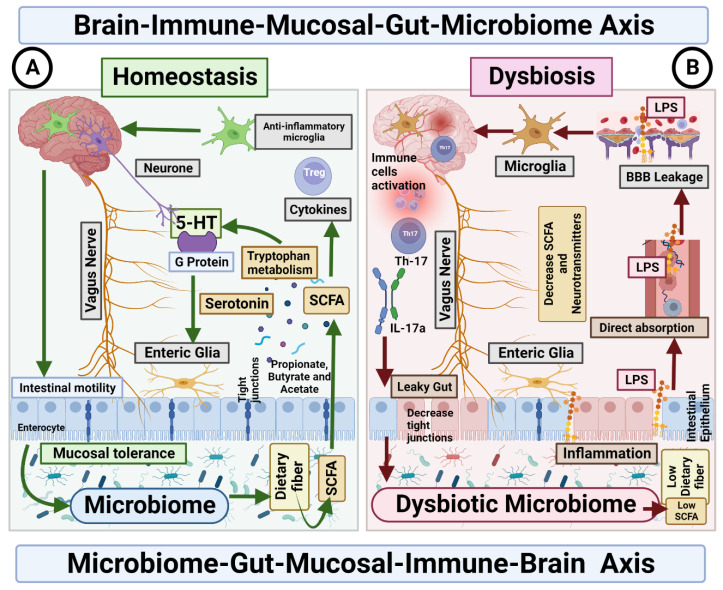
**Microbiome**–**Gut**–**Mucosal**–**Immune**–**Brain bidirectional relationship.** The vagus nerve transfers information on the state of the digestive system to the brain through sensory fibres. It sends neuronal, endocrine and immune signals. The gut microbiota metabolites are involved in permeability of the intestinal epithelium, neurotransmitter synthesis and cytokines release. (**A**) **Homeostasis**. Under homeostasis conditions, gut microbiome and diet control the activity of ENS cells through SCFA, which in turn activate the enteroendocrine cells of the intestinal epithelium, which increases the production of serotonin and therefore changes intestinal motility. In addition, the microbiome is essential for the maintenance of mucosal glial cells that express neurotrophic factor. (**B**) **Dysbiosis**. The imbalance in the homeostasis of the gut microbiome is associated with the decreased production of metabolites, such as butyrate, responsible for the modulation of inhibitory and excitatory pathways, and with anti-inflammatory effects on microglia. Deficiency of this microbial metabolite could be the cause of a disruption in the intercommunication between the ENS and the mucosal immune system.

**Table 1 behavsci-13-00548-t001:** Neurotransmitter synthesis and release is altered by the dysbiotic gut microbiome in ASD.

Neurotransmitters *	Taxon Name of Involved Microorganism	Effect on Neurotransmitter Level in Dysbiosis in ASD	Role of Microorganisms in the Human Body	References
GABA/Acetylcholine Noradrenaline (norepinephrine)/Dopamine	(g) *Lactobacillus* sp.^1^	Decrease ↓	Improve the brain function and elevated mood.	[38,40,41]
GABA	(g) *Bifidobacterium* ^1^	Decrease ↓	Regulates emotions and behaviour. Maintains gut homeostasis, produces vitamins and antimicrobial substances and regulates the host immune system.	[32,38,41]
Noradrenaline (norepinephrine)/Serotonin	(g) *Escherichia* ^1^	Decrease ↓	Produces active molecules that may reach and influence the CNS after the secretion into the periphery or by activating afferent neurons.	[12,38]
Noradrenaline (norepinephrine)	(g) *Saccharomyces* sp.^2^	Increase ↑	Involved in ASD pathogenesis through immune factors and may play an essential role in the development of ASD.	[38,42]
Serotonin	(g) *Candida* ^2^	Increase ↑	In a dysbiotic environment as frequently observed in the autistic population, the yeast proliferates and produces ammonia and toxins, which increase autistic behaviour.	[12,38,43]
	(g) *Streptococcus* sp.^1^	Decrease ↓	Protects tissues from oxidative stress.	[12,38,41]
	(g) *Enterococcus* sp.^1^	Decrease ↓	DNA damage in colorectal cancer.	[12,38,41]
Tryptophan	(g) *Clostridium^1^*	Increase ↑	Increases the production of antioxidant and neuro-protectant molecules inside the gut; acts as a biomarker for ASD; inhibits the growth of other gut microbiota, promotes the growth or virulence of gut pathogens.	[41,44]

^1^ Bacteria. ^2^ Fungi. (g): genus. ↑ Increase. ↓ Decrease. * These neurotransmitters produced by bacteria and fungi can cross the intestinal barrier, and potentially play a role in regulating brain functions (neurogenesis, neurodevelopment, mental homeostasis, background emotions and immune response modulation).

## Data Availability

Not applicable.

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
