# Peer review of "Microbiome–Gut–Mucosal–Immune–Brain Axis and Autism Spectrum Disorder (ASD): A Novel Proposal of the Role of the Gut Microbiome in ASD Aetiology"

_behavsci, 2023, doi:10.3390/bs13070548_

Round 1

Reviewer 1 Report

In this review article, the authors are trying to give an account on the association of microbiome- gut- mucosal- immune-brain axis and ASD, and which is an important area of research to explore. The article is well written, but it is very general. Authors reported many earlier studies in this review, but the specific research question that the authors needed to explore through this review article needed more attention. Although they have mentioned their proposal in the Figure 2, it is very brief. In a review article, authors need to address and elaborate their specific interest and their recommendations, and which should be the priority for an experimental study. This article is more like an account of what others have done in this field.

Minor comments- In page 2, first paragraph, under introduction, line 45-49, the sentence needed to be reconstructed.

Author Response

Comment:

In this review article, the authors are trying to give an account on the association of microbiome- gut- mucosal- immune-brain axis and ASD, and which is an important area of research to explore. The article is well written, but it is very general. Authors reported many earlier studies in this review, but the specific research question that the authors needed to explore through this review article needed more attention. Although they have mentioned their proposal in the Figure 2, it is very brief. In a review article, authors need to address and elaborate their specific interest and their recommendations, and which should be the priority for an experimental study. This article is more like an account of what others have done in this field.

Response to reviewer:

In this revised version of the manuscript, we have incorporated additional recent studies and reports that pertain to each of the various topics discussed in the text. We have made an effort to include more up-to-date information that we believe will be valuable to the readers.

With respect to our proposal on the involvement of gut microbiota in ASD, as presented in figures one and two, we have enhanced the information and further elucidated the foundations that support this model. The section has been thoroughly rewritten, and you will find the modifications highlighted in red in the revised version.

ASD can be syndromic or non-syndromic. Syndromic ASD is often associated with chromosomal abnormalities or monogenic alterations such as Rett and Fragile X syndrome among other syndromes. Non-syndromic ASD is a complex and multifactorial condition influenced by both genetic and environmental factors (119, 134, 135, 136, 137, 138). It is widely recognized that the causative traits of ASD predominantly have a genetic basis. A meta-analysis of 13 twin studies estimated the heritability of ASD to be 74%, with an environmental effect of 25% (147). However, it is important to acknowledge that the heritability value may be prone to overestimation due to statistical artifacts or an underestimation of unaccounted shared environmental factors, which cannot be accurately estimated in twin and sibling studies. Another estimation of ASD heritability based on 192 twin pairs found a lower value of 38% for heritability and a 58% contribution from environmental factors (148).

The contribution of genetic and environmental factors to the presentation of ASD can vary, but genetic factors are considered to be the primary determinants. Approximately 1000 human loci have been associated with ASD according to SFARI database (149). Additionally, a meta-analysis of genome-wide association studies (GWAS) involving over 16,000 individuals identified chromosomal regions 10q24.32, 3p13, 3p25, and 8p11.23 that are associated with ASD. It is worth noting that the 10q24.32 region is also linked to schizophrenia (150).

In terms of environmental factors, advanced parental age (151), prenatal exposure to antibiotics or certain drugs (152), maternal immune activation (153), imbalances in prenatal micronutrients (154), epilepsy (155, 156), and maternal obesity (157) have been shown to be associated with an increased risk of ASD. Therefore, these factors are considered as potential environmental drivers in the aetiology of ASD.

While there is a wealth of evidence suggesting a potential involvement of the gut microbiota in the manifestation of core ASD symptoms, it is important to note that the precise role of gut microbiota in the pathogenesis of ASD remains unclear. Numerous studies have focused on animal models, such as germ-free mice, while others have analysed dysbiosis in ASD patients and its impact on immunoregulators, neurotransmitters, and relevant metabolite production in ASD (93, 101, 102, 103). However, despite the available information, no study to date has definitively established the intestinal microbiota as a causative environmental factor in ASD. Nevertheless, it has been a therapeutic target in various interventions aimed at managing ASD symptoms in patients.

The relevance of dysbiosis as a factor in the aetiology of ASD relies on the fact that it is modifiable. There are reports of interventions that have had beneficial but modest effects on autistic manifestations. For example, faecal transplant therapies and probiotic supplementation. A pioneer group in faecal transplantation in children with ASD found that after ten weeks of intervention, there was a reduction of almost 80% in gastrointestinal symptoms and improvement in behavioural symptoms, and that these improvements persisted after eight weeks of treatment; in addition, there were beneficial changes in the abundance of Bifidubacterium, Prevotella, and Desulfovibrio; these changes also persisted after the suspension of the intervention and two years later (141, 142). Regarding probiotics, a successful study that supplemented with probiotics children with ASD for 6 months found positive effects on some gastrointestinal symptoms, adaptive functions, and sensory profiles versus the placebo group (143). Another study found that after supplementation with probiotics and prebiotics, there was an increase in beneficial bacteria (Bifidubacterium and B. longum) and suppression of pathogenic bacteria (Clostridium) with a significant reduction in the severity of autistic and gastrointestinal symptoms (144).

Dysbiosis has been proposed as a potential contributing factor in the development of ASD. However, its relevance in ASD aetiology remains unclear due to inconclusive results from various studies. Some studies suggest a possible association between dysbiosis and autistic symptoms (31, 60), while others dismiss the notion that dysbiosis plays a decisive role in the development of ASD (19, 140).

Yap and colleagues proposed an alternative perspective, suggesting that gut dysbiosis observed in ASD patients could be a consequence of the restrictive and behavioural patterns inherent to ASD. They argue that alterations in eating patterns, rather than dysbiosis, might have a minor or non-causal role in ASD. This study highlights the unconvincing evidence supporting the participation of gut microbiota in ASD. Many studies lacked statistical power due to small sample sizes and failed to consider important confounders such as sex, stool consistency, and diet (19).

In this study, we present a novel model that incorporates the role of gut microbiota in ASD while considering the findings presented by Yap and colleagues aims to explain the limited but evident effects of microbiota intervention on ASD core symptoms in patients (Figure 1).

According to our model, the primary driving factors in the aetiology of ASD are genetic and environmental, with no direct involvement of gut microbiota in the manifestation of ASD. However, we propose that the restricted feeding patterns observed in ASD patients lead to decreased food diversity, subsequently resulting in a reduction of microbiota diversity and the development of gut dysbiosis.

The dysbiosis of the gut microbiota affects the gut-brain axis through mechanisms discussed in detail in this review (Figure 2). Therefore, the severity of autistic manifestations is accentuated. Within our proposed model, gut dysbiosis acts as a secondary factor that exacerbates the core symptoms of ASD and contributes to the occurrence of gastrointestinal manifestations until a certain threshold of severity is reached.

By incorporating the role of gut dysbiosis in accentuating ASD symptoms, our model provides a comprehensive framework that integrates genetic and environmental factors with the influence of the gut microbiota. This holistic perspective allows for a better understanding of the complex interactions underlying ASD pathogenesis and highlights the potential importance of interventions targeting gut dysbiosis in the management of ASD symptoms.

Therefore, based on the proposed model (Figure 1 and Figure 2), intervening in the vicious cycle by restoring the gut microbiota, specifically by increasing the abundance of beneficial bacteria, in individuals with ASD may lead to a modest improvement in autistic manifestations, ultimately reaching the severity level predetermined by the underlying etiologic environmental and genetic factors of ASD. Additionally, such interventions have been reported to alleviate gastrointestinal manifestations, as demonstrated in intervention studies (143, 144).

By targeting the dysbiosis of the gut microbiota and promoting a healthier microbial composition, interventions hold the potential to positively influence ASD symptoms. However, it is important to note that the extent of improvement may be limited due to the dominant influence of genetic and environmental factors on the overall severity of ASD.

These findings suggest that therapeutic approaches aimed at restoring a balanced gut microbiota profile may provide some benefits for individuals with ASD, particularly in terms of gastrointestinal symptoms. Nonetheless, it is crucial to recognize that microbiota-based interventions should be considered as complementary to comprehensive treatment strategies that address the multifaceted nature of ASD. Further research is warranted to explore the optimal interventions and long-term effects on the gut-brain axis in individuals with ASD.

Minor comments- In page 2, first paragraph, under introduction, line 45-49, the sentence needed to be reconstructed.

The section was rewritten, and you can find the changes in red in the revised version.

During the first 3 years of life, the consolidation of the microbiota-gut-brain axis is of vital importance because during this stage the Central Nervous System (CNS) develops rapidly (7), and gut microbiota also suffer important changes due to changes in feeding patterns during this developmental stage (8, 9). 

Reviewer 2 Report

The present review related to Microbiome-Gut-Mucosal-Immune-Brain Axis and Autism Spectrum Disorder (ASD): A Novel Proposal of the Role of Gut Microbiome in ASD Aetiology is interesting, full of information and overall well-designed. However, edits must be made before being considered for publication in Behavioral Sciences.

 Edits and suggestions :

 1.     In the abstract, line 14, it would be more precise to indicate "deficit social interaction and communication", and "Restricted and stereotyped behaviors, interests or activities".

 2.     In the keywords, it would be better to note Autism Spectrum Disorder instead of ASD.

 3.     Introduction should be enriched by some recent references (2020-2023).

 4.     In introduction, could you explain more what are the factors/causes that create dysbiosis.

 5.      L.75-76 "The resident microbial community of the human gut has a unique combination of different types and numbers of bacteria that are constantly evolving and can be influenced  by diet, environmental factors, drug treatments, and age, and between many other factors", it would be interesting to briefly develop each of them.

 6.      L.84-86 "Dysbiosis is generally associated with pathological conditions such as
obesity, type 2 diabetes, cancer, gastrointestinal diseases, and neuropsychiatric and neurodevelopmental diseases", could you specify what type of neuropsychiatric and neurodevelopmental diseases.

 7.      L.109-111 "The gut microbiota also regulates central neurotransmitters by altering the levels of their precursors (Table 1)”, please specify in certain disorders, in particular neurodevelopmental disorders such as ASD or put table 1 in the part 2.4.

 8.      You could insert figure 1 below line 488.

 9.      In figure 1, please specify that there are several possible environmental causes not only "advanced maternal age and antibiotic use during pregnancy" and in the text it would also be interesting for the reader to briefly explain the genetic and environmental causes of ASD.

  10.   L.530-531 "Alterations in the microbiota-gut-brain axis are the main cause of the most frequent gastrointestinal motility disorders in psychiatric and neurodevelopmental diseases", please specify what type of psychiatric and neurodevelopmental diseases.

11.   L.531-533 "Neuroinflammation produced by the different metabolites of a dysbiotic microbiota may be a pathogenic factor in severe neurodegenerative disorders", please specify what type of neurodegenerative diseases.

 12.  In the conclusion, it would be relevant and useful to further develop preventive and therapeutic perspectives in ASD indicating some recent studies.

Minor editing of English language required

Author Response

The present review related to Microbiome-Gut-Mucosal-Immune-Brain Axis and Autism Spectrum Disorder (ASD): A Novel Proposal of the Role of Gut Microbiome in ASD Aetiology is interesting, full of information and overall well-designed. However, edits must be made before being considered for publication in Behavioral Sciences.

Edits and suggestions:

  1. In the abstract, line 14, it would be more precise to indicate "deficit social interaction and communication", and "Restricted and stereotyped behaviors, interests or activities".

We made the change, and changed the same recommendation in the Introduction L 34-35. All the changes were highlighted in red.

  1. In the keywords, it would be better to note Autism Spectrum Disorder instead of ASD.

We made the change.

  1. Introduction should be enriched by some recent references (2020-2023).

We agree, this is a relevant point and have included some recent references (2020-2023): 3, 4, 5, 6, 7, 9, 11, 12, 14, 15 and 17.

  1. In introduction, could you explain more what are the factors/causes that create dysbiosis.

We have explained more about factors/causes that create dysbiosis in the introduction. The next text was introduced and highlighted in red:

The mode of birth and feeding type during the first years of life can condition changes in the gut microbiota. Children delivery vaginally and breastmilk-fed presented more Bifidobacterium than children delivery by caesarean section and formula-fed. In the complementary feeding stage, the diversity of foods can determine the conformation of different microbial groups that will be maintained in the adult stage. A long-term fibre-deficient diet and high levels of antibiotic treatments (≥4 courses) can lead to changes in gut microbiota diversity and structure (dysbiosis). These changes cause alterations in the production of important neurotransmitters for neurodevelopment (5).”

  1. 75-76 "The resident microbial community of the human gut has a unique combination of different types and numbers of bacteria that are constantly evolving and can be influenced  by diet, environmental factors, drug treatments, and age, and between many other factors", it would be interesting to briefly develop each of them.

We develop about factors that affect the gut microbiota and introduced the next text highlighted in red:

For example, the western-style diet (high fat/low fibre) is related to the decrease in Bacteroides, Bifidobacterium, and Akkermansia genus, affecting intestinal microbiota homeostasis and permeability. Furthermore, the western diet is characterized by an increased Firmicutes/Bacteroidetes ratio and weight gain. A relationship between obesity and the risk of infection by the enteric pathogen Clostridium difficile have been identified (146). Antibiotic exposure during infancy is associated with adiposity and obesity later in childhood, which may occur through disrupted microbial colonization that can have consequent effects on gastrointestinal motility, secretion, and immune functions. Polypharmacy in older hospitalized patients is associated with dysbiosis in contrast to no dysbiosis in active healthy older people without polypharmacy. In terms of lifestyle, changes in living conditions and dietary changes have been shown to significantly influence the elderly gut microbiota (147). In healthy individuals, the effect of geographic distance has an impact on gut microbiota variations, demonstrating the cumulative effect of dietary, genetic, lifestyle, and climate factors on microbiota variations among healthy individuals (25).

  1. 84-86 "Dysbiosis is generally associated with pathological conditions such as
    obesity, type 2 diabetes, cancer, gastrointestinal diseases, and neuropsychiatric and neurodevelopmental diseases", could you specify what type of neuropsychiatric and neurodevelopmental diseases.

We changed the text for the next:

Dysbiosis is generally associated with pathological conditions such as obesity, type 2 diabetes, cancer, gastrointestinal diseases, and neuropsychiatric as depression and anxiety, and also neurodevelopmental diseases as ASD (5, 21).”

  1. 109-111 "The gut microbiota also regulates central neurotransmitters by altering the levels of their precursors (Table 1)”, please specify in certain disorders, in particular neurodevelopmental disorders such as ASD or put table 1 in the part 2.4.

We specified disorders in the text

The gut microbiota also regulates central neurotransmitters by altering the levels of their precursors in ASD, depression and anxiety, in particular neurodevelopmental disorders such as ASD (Table 1)”

  1. You could insert figure 1 below line 488.

We changed the position of figure 1.

  1. In figure 1, please specify that there are several possible environmental causes not only "advanced maternal age and antibiotic use during pregnancy" and in the text it would also be interesting for the reader to briefly explain the genetic and environmental causes of ASD.

We changed in figure 1:

  1. SFARI (Simons Foundation Autism Research Initiative) gene database, lists about 1000 genes associated with ASD. And 2. Advanced maternal age; antibiotic or drugs use during pregnancy; maternal immune activation, prenatal imbalances of micronutrients, and maternal gastrointestinal disorders, epilepsy, and obesity. (Cambiar en figura 1)

In the text as a recommendation of reviewer 1 we made major changes in the section were we deepen in the etiological factors in ASD, please revise the changes that are highlighted in red.

ASD can be syndromic or non-syndromic. Syndromic ASD is often associated with chromosomal abnormalities or monogenic alterations such as Rett and Fragile X syndrome among other syndromes. Non-syndromic ASD is a complex and multifactorial condition influenced by both genetic and environmental factors (119, 134, 135, 136, 137, 138). It is widely recognized that the causative traits of ASD predominantly have a genetic basis. A meta-analysis of 13 twin studies estimated the heritability of ASD to be 74%, with an environmental effect of 25% (147). However, it is important to acknowledge that the heritability value may be prone to overestimation due to statistical artifacts or an underestimation of unaccounted shared environmental factors, which cannot be accurately estimated in twin and sibling studies. Another estimation of ASD heritability based on 192 twin pairs found a lower value of 38% for heritability and a 58% contribution from environmental factors (148).

The contribution of genetic and environmental factors to the presentation of ASD can vary, but genetic factors are considered to be the primary determinants. Approximately 1000 human loci have been associated with ASD according to SFARI database (149). Additionally, a meta-analysis of genome-wide association studies (GWAS) involving over 16,000 individuals identified chromosomal regions 10q24.32, 3p13, 3p25, and 8p11.23 that are associated with ASD. It is worth noting that the 10q24.32 region is also linked to schizophrenia (150).

In terms of environmental factors, advanced parental age (151), prenatal exposure to antibiotics or certain drugs (152), maternal immune activation (153), imbalances in prenatal micronutrients (154), epilepsy (155, 156), and maternal obesity (157) have been shown to be associated with an increased risk of ASD. Therefore, these factors are considered as potential environmental drivers in the aetiology of ASD.

While there is a wealth of evidence suggesting a potential involvement of the gut microbiota in the manifestation of core ASD symptoms, it is important to note that the precise role of gut microbiota in the pathogenesis of ASD remains unclear. Numerous studies have focused on animal models, such as germ-free mice, while others have analysed dysbiosis in ASD patients and its impact on immunoregulators, neurotransmitters, and relevant metabolite production in ASD (93, 101, 102, 103). However, despite the available information, no study to date has definitively established the intestinal microbiota as a causative environmental factor in ASD. Nevertheless, it has been a therapeutic target in various interventions aimed at managing ASD symptoms in patients.

The relevance of dysbiosis as a factor in the aetiology of ASD relies on the fact that it is modifiable. There are reports of interventions that have had beneficial but modest effects on autistic manifestations. For example, faecal transplant therapies and probiotic supplementation. A pioneer group in faecal transplantation in children with ASD found that after ten weeks of intervention, there was a reduction of almost 80% in gastrointestinal symptoms and improvement in behavioural symptoms, and that these improvements persisted after eight weeks of treatment; in addition, there were beneficial changes in the abundance of Bifidubacterium, Prevotella, and Desulfovibrio; these changes also persisted after the suspension of the intervention and two years later (141, 142). Regarding probiotics, a successful study that supplemented with probiotics children with ASD for 6 months found positive effects on some gastrointestinal symptoms, adaptive functions, and sensory profiles versus the placebo group (143). Another study found that after supplementation with probiotics and prebiotics, there was an increase in beneficial bacteria (Bifidubacterium and B. longum) and suppression of pathogenic bacteria (Clostridium) with a significant reduction in the severity of autistic and gastrointestinal symptoms (144).

Dysbiosis has been proposed as a potential contributing factor in the development of ASD. However, its relevance in ASD aetiology remains unclear due to inconclusive results from various studies. Some studies suggest a possible association between dysbiosis and autistic symptoms (31, 60), while others dismiss the notion that dysbiosis plays a decisive role in the development of ASD (19, 140).

Yap and colleagues proposed an alternative perspective, suggesting that gut dysbiosis observed in ASD patients could be a consequence of the restrictive and behavioural patterns inherent to ASD. They argue that alterations in eating patterns, rather than dysbiosis, might have a minor or non-causal role in ASD. This study highlights the unconvincing evidence supporting the participation of gut microbiota in ASD. Many studies lacked statistical power due to small sample sizes and failed to consider important confounders such as sex, stool consistency, and diet (19).

In this study, we present a novel model that incorporates the role of gut microbiota in ASD while considering the findings presented by Yap and colleagues aims to explain the limited but evident effects of microbiota intervention on ASD core symptoms in patients (Figure 1).

  1. 530-531 "Alterations in the microbiota-gut-brain axis are the main cause of the most frequent gastrointestinal motility disorders in psychiatric and neurodevelopmental diseases", please specify what type of psychiatric and neurodevelopmental diseases.

We changed the sentence for: “Alterations in the microbiota-gut-brain axis are the main cause of the most frequent gastrointestinal motility disorders in psychiatric such as depression and anxiety, and neurodevelopmental diseases as ASD.”

  1. L.531-533 "Neuroinflammation produced by the different metabolites of a dysbiotic microbiota may be a pathogenic factor in severe neurodegenerative disorders", please specify what type of neurodegenerative diseases.

We changed the sentence for: “Neuroinflammation produced by the different metabolites of a dysbiotic microbiota may be a pathogenic factor in severe neurodegenerative disorders such as Alzheimer’s disease and Parkinson’s disease.”

  1. In the conclusion, it would be relevant and useful to further develop preventive and therapeutic perspectives in ASD indicating some recent studies.

We added the next text in conclusions:

 The intercommunication between immune-mediated neurological and gastrointestinal disorders, leading to the proposition of a novel microbiome-gut-mucosa-immuno-brain axis, termed neuroimmunogastroenterology, in individuals with ASD. We postulate that this axis plays a significant role in a vicious cycle that contributes to the exacerbation of autistic manifestations. To gain insights into this phenomenon, large-scale clinical cohort studies involving early diagnosed ASD patients are essential to assess the potential prevention of dysbiosis and avoidance of accentuated autistic behaviour. Identifying therapeutic interventions to restore a healthy gut microbiome, thereby alleviating autistic manifestations, is paramount. Strategies such as diversifying food intake with sensory feeding therapy, along with targeted administration of development stage-specific pre and probiotics to restore beneficial intestinal bacteria, are potential approaches. Furthermore, exploring the promising long-term positive effects of faecal microbiome transplantation is warranted.

Round 2

Reviewer 1 Report

Authors addressed the specific concerns pointed by the initial review. Certain minor things need to be addressed, such as the following,

Importantly, in the page 12, line 556-560 authors mention, in their starting sentence  “In this study, we present a novel model that incorporates the role of gut microbiota in ASD while considering the findings presented by Yap and colleagues aims…..”, since this is a review article and there is no studies in this manuscript, the authors are only proposing their hypothesis and that will lead to a research study through this review article. So, the authors should consider that in mind and rewrite these sentences accordingly in this article.    

Author Response

Reviewer comment:

Importantly, in the page 12, line 556-560 authors mention, in their starting sentence  “In this study, we present a novel model that incorporates the role of gut microbiota in ASD while considering the findings presented by Yap and colleagues aims…..”, since this is a review article and there is no studies in this manuscript, the authors are only proposing their hypothesis and that will lead to a research study through this review article. So, the authors should consider that in mind and rewrite these sentences accordingly in this article.   

Answer: We do agree that it was a redaction mistake, que change the paragraph to the next one:

In this review, we propose a novel model that incorporates the role of gut microbiota in ASD while considering the findings presented by Yap and colleagues aims to explain the limited but evident effects of microbiota intervention on ASD core symptoms in patients (Figure 1).”

The changes were highlighted in red.

Reviewer 2 Report

The authors have considered and responded to all my edits and suggestions. I accept this paper in its current form.

Author Response

Thank you for your previous comments, we consider that were very useful to improve the manuscript.